# 1,25-Dihydroxyvitamin D_3_ Provides Benefits in Vitiligo Based on Modulation of CD8+ T Cell Glycolysis and Function

**DOI:** 10.3390/nu15214697

**Published:** 2023-11-06

**Authors:** Yujia Wei, Tingmei Wang, Xiaoqi Nie, Zeqi Shi, Zhong Liu, Ying Zeng, Ronghua Pan, Ri Zhang, Yunhua Deng, Dong Li

**Affiliations:** Department of Dermatology, Tongji Hospital, Tongji Medical College, Huazhong University of Science and Technology, Wuhan 430030, China; m202176167@hust.edu.cn (Y.W.); dwtm198935@126.com (T.W.); nxq1625712361@163.com (X.N.); zeqishi_2022@tjh.tjmu.edu.cn (Z.S.); doctorliuz@126.com (Z.L.); cnzengying@163.com (Y.Z.); m202076193@hust.edu.cn (R.P.); zr080130@163.com (R.Z.); yhdeng@hust.edu.cn (Y.D.)

**Keywords:** vitamin D, vitiligo, CD8+ T cells, glycolysis

## Abstract

Vitiligo is a common autoimmune skin disease caused by autoreactive CD8+ T cells. The diverse effects of 1,25-dihydroxyvitamin D₃ [1,25(OH)₂D₃] on immune cell metabolism and proliferation have made it an interesting candidate as a supporting therapeutic option in various autoimmune diseases. This study aimed to elucidate the immunomodulatory effects of 1,25(OH)₂D₃ in vitiligo. Cross-sectional relationships between serum 1,25(OH)₂D₃ levels and disease characteristics were investigated in 327 patients with vitiligo. The immunomodulatory and therapeutic effects of 1,25(OH)₂D₃ were then investigated in vivo and in vitro, respectively. We found that 1,25(OH)₂D₃ deficiency was associated with hyperactivity of CD8+ T cells in the vitiligo cohort. In addition, 1,25(OH)₂D₃ suppressed glycolysis by activating the AMP-activated protein kinase (AMPK) signaling pathway, thereby inhibiting the proliferation, cytotoxicity and aberrant activation of CD8+ T cells. Finally, the in vivo administration of 1,25(OH)₂D₃ to melanocyte-associated vitiligo (MAV) mice reduced the infiltration and function of CD8+ T cells and promoted repigmentation. In conclusion, 1,25(OH)₂D₃ may serve as an essential biomarker of the progression and severity of vitiligo. The modulation of autoreactive CD8+ T cell function and glycolysis by 1,25(OH)₂D₃ may be a novel approach for treating vitiligo.

## 1. Introduction

Vitiligo is a common depigmentation disease characterized by the progressive destruction of melanocytes, resulting in chalky-white macules. Multiple hypotheses of melanocyte destruction in vitiligo have been proposed, including genetics, oxidative stress, production of inflammatory mediators, melanocyte detachment mechanisms and, most importantly, autoimmune reactions [1]. Among all these hypotheses, autoreactive cytotoxic CD8+ T cells have been shown to be most necessary for the elimination of melanocytes and for promoting the progression of vitiligo [2]. However, regarding treatment, there is a limited number of options, including topical tacrolimus and steroids, phototherapy, Janus kinase (JAK) inhibitors and surgical procedures, and this disease is frequently refractory to all existing treatment modalities [3]. Effective therapies targeting pathogenic autoreactive CD8+ T cells need to be proposed and investigated.

Vitamin D is an essential hormone synthesized in the skin and is responsible for skin pigmentation [4]. The metabolite of vitamin D, 1,25-dihydroxyvitamin D_3_ [1,25(OH)₂D₃], produced via two hydroxylation steps, has better bioavailability and not only exhibits anti-proliferative and pro-differentiation effects but also acts as an immunomodulator in a range of immune cell lines [5]. Observational studies based on the concentration of 1,25(OH)₂D₃ revealed its beneficial effects in a variety of autoimmune diseases [6]. However, vitamin D deficiency has become a common complication encountered in daily clinical practice with a prevalence range from 2 to 30 % across Europe [4]. Although the role of vitamin D deficiency in numerous autoimmune diseases is well established, the relationship between vitamin D levels and vitiligo has not yet been fully elucidated. Not only are there few studies investigating the relationship between vitiligo and reduced levels of vitamin D, but those few have provided conflicting results. For instance, a case–control study found that circulating 1,25(OH)₂D₃ levels were lower in patients with vitiligo than in matched healthy controls [7]. Similarly, a pilot study found that more than 50% of patients had insufficient levels of 1,25(OH)₂D₃, which suggested that vitamin D deficiency may be a risk factor for vitiligo development [8]. However, another case–control study found no significant difference in serum vitamin D levels between vitiligo patients and matched controls [9]. Furthermore, a recent study using Mendelian randomization analysis showed no causal effect of circulating 1,25(OH)₂D₃ on the risk of vitiligo [10]. These inconsistent results may be due to differences in study populations, sample sizes and measurement methods, as well as biases such as confounders and reverse causality. It is still unknown if vitamin D deficiency plays a role in vitiligo, as it does in other autoimmune diseases.

Recent studies of vitiligo indicate that IFN-γ signaling produced by cytotoxic CD8+ T cells that target epidermal melanocytes is essential for depigmentation of the skin [11]. Upon encountering antigens, T cells shift to glycolytic metabolism to maintain effector functions. High glycolytic flux is accompanied by upregulation of the expression of components of the effector machinery, including IFN-γ [12]. Until now, it was assumed that CD4+ T cells were the typical target cell population for the immunomodulatory effects of 1,25(OH)₂D₃. The inhibition of aerobic glycolysis by 1,25(OH)₂D₃ substantially contributes to its immune-regulatory capacity in CD4+ T cells and significantly blunts the expression of IFN-γ, whereas the effects of this hormone on CD8+ lymphocytes have rarely been reported [13,14]. Since endogenous autoreactive CD8+ T cells are the main pathogenic cells in vitiligo, whether 1,25(OH)₂D₃ is involved in regulating human CD8+ T cell function and in vitiligo pathogenesis still needs to be more thoroughly investigated.

As vitamin D deficiency is a risk factor for various autoimmune diseases, the present study evaluates whether low serum 1,25(OH)₂D₃ levels also play a pathogenic role in vitiligo. In addition to the initiation and progression of vitiligo, the IFN-γ released by CD8+ T cells is also required to maintain established lesions, positioning the secretion of IFN-γ by CD8+ T cells as a potential therapeutic target [2]. To ascertain whether 1,25(OH)₂D₃ could prevent vitiligo induction via a mechanism involving the secretion of IFN-γ by CD8+ T cells, we conducted a cross-sectional study in which the clinical data and specimens were collected from patients with vitiligo. We then investigated the regulation of 1,25(OH)₂D₃ on glucose metabolism and immunity in CD8+ T cells as well as related signaling pathways in vitro. Further, a mouse model of vitiligo based on the activation of endogenous autoreactive CD8+ T cells targeting epidermal melanocytes was used to validate the therapeutic effect of 1,25(OH)₂D₃ [15].

## 2. Materials and Methods

### 2.1. Subjects

In this cross-sectional study, 327 patients with vitiligo were enrolled from the dermatology clinic of Tongji Hospital, Tongji Medical College of Huazhong University of Science & Technology, between February 2022 and February 2023. For every patient, a complete history was taken and a detailed cutaneous examination was performed. Vitiligo was diagnosed based on patient history and was classified into segmental, nonsegmental or unclassified/undetermined vitiligo depending upon the clinical presentation [16]. Patients with a history of phototherapy, vitamin D supplementation and other autoimmune diseases were excluded from the study. Vitiligo activity was evaluated using the history-based Vitiligo Disease Activity (VIDA) score, a six-point scale for assessing vitiligo stability over time. It depends on the patient’s report for disease activity. Vitiligo extent and pigmentation were evaluated using Vitiligo Area Severity Index (VASI) scores. The extent of vitiligo as per hand units is multiplied by the extent of depigmentation in different body sites, which are added together to obtain the total VASI score [17]. Serum 1,25(OH)₂D₃ concentration was measured as specified in the procedure, and it was confirmed that there were no significant changes in lifestyle to minimize the impact of sunlight on serum 1,25(OH)₂D₃. All samples were tested by the same laboratory and at the same time.

The peripheral blood samples were collected from patients with vitiligo (*n* = 6) and healthy controls (*n* = 6) into a vacutainer tube containing heparin. All patients provided written informed consent. Ethical clearance was obtained from the ethical review board of Tongji Hospital (TJ-IRB20220150).

### 2.2. Procedure

#### 2.2.1. Cell Culture and Treatment

Peripheral blood mononuclear cells (PBMCs) were isolated using lymphocyte cell separation media (Solarbio, Beijing, China) following the manufacturer’s instructions. For CD8+ T cell isolation and activation, CD8+ T cells were magnet-based and separated from PBMCs of patients with vitiligo and controls using a MojoSort™ Human CD8 T Cell Isolation Kit (BioLegend, San Diego, CA, USA). The procedures were performed based on instructions provided in the kit manual. The purity of separated CD8+ T cells was typically >95%. CD8+ T cells were cultured in 96-well round-bottom plates and activated using a Streptamer^®^ CD3/CD28 kit for T cell expansion (Iba-lifescience, Goettingen, Germany) according to the manufacturer’s instructions. After activation, CD8+ T cells were then resuspended at 1 × 106/mL in RPMI-1640 (Thermo Fisher Scientific, Waltham, MA, USA) containing 10% fetal calf serum (Gibco, Grand Island, NY, USA), 50 U/mL penicillin, 50 mg/mL streptomycin (Thermo Fisher Scientific, Waltham, MA, USA) and 50 IU/mL rIL-2 (Thermo Fisher Scientific, Waltham, MA, USA). In experiments requiring low-glucose media, CD8+ T cells were cultured in RPMI-1640 without glucose (Pricella, Wuhan, China) containing 10% FCS, 50 U/mL penicillin, 50 mg/mL streptomycin, 50 IU/mL rIL-2 and 5 mM D-(+)-Glucose (Sigma-Aldrich, Shanghai, China). All treatments were added at the beginning and remained present for the duration of culture. The treatments associated with each group are detailed in Figure 1. For the treatment with 1,25(OH)₂D₃ (MedChemExpress, Shanghai, China), it was reconstituted in ethanol, stored in concentrated solutions at −80 °C and used rapidly on ice and out of light to avoid concentration variations due to ethanol evaporation. 1,25(OH)₂D₃ was freshly diluted with 0.9% NaCl before each experiment. Primary cultures of normal human epidermal melanocytes (NHEMs) were isolated and cultured as our previous experiments described [18]. Activated CD8+ T cells (2 × 105/well) were cultured with NHEMs for 12 h in 96-well flat-bottom plates after being pretreated with 1,25(OH)₂D₃ at concentrations of 0 nM, 10 nM and 100 nM for 72 h, respectively. NHEMs treated with 10 nM and 100 nM of 1,25(OH)₂D₃ for 12 h were used as negative controls. Apoptosis was detected using flow cytometry with Annexin V/PI staining. TALL-104 cells (ATCC, Manassas, VA, USA) were maintained in RPMI-1640 (Thermo Fisher Scientific, Waltham, MA, USA) supplemented with 10% fetal calf serum (Gibco, Grand Island, NY, USA) and 50 IU/mL rIL-2 (Thermo Fisher Scientific, Waltham, MA, USA). Compound C (MedChemExpress, Shanghai, China), a potent AMPK inhibitor, was dissolved in dimethyl sulfoxide. For protein inhibition, Tall-104 cells were first pretreated with 10 um Compound C for 18 h, and then with 100 nM 1,25(OH)₂D₃ for 72 h. B16F10 cells (ATCC, Manassas, VA, USA) were maintained in DMEM medium supplemented with 10% FCS, 50 U/mL penicillin and 50 mg/mL streptomycin.

#### 2.2.2. Animals and Drug Treatment Protocols

Adult male C57BL/6 mice (18–22 g, five weeks old) were procured from Beijing Vital River Laboratory Animal Technology (Beijing, China). All mice were socially housed under a 12 h light/dark cycle (lights on at 8:00 AM) with free access to food and water, with a constant temperature of 24 ± 1 °C and 50–56% humidity.

Mice were randomly divided into two groups: control mice (*n* = 5) and mice subjected to vitiligo induction (*n* = 15). Recent studies have shown that melanoma immunotherapy is associated with the development of vitiligo in patients with melanoma via the activation of CD8+ T cells that recognize antigens shared by melanoma cells and melanocytes [19]. The significant reduction in Tregs in patients with vitiligo was also proposed as a main mechanism leading to endogenous autoreactive CD8+ T cell overactivation and inducing vitiligo in patients [20]. Based on the above mechanism, a new mouse model of melanoma-Treg-induced vitiligo was established recently [11]. This mouse model of melanocyte-associated vitiligo (MAV) was constructed by activating endogenous autoreactive CD8+ T cells targeting epidermal melanocytes, which could efficiently induce vitiligo in any C57BL/6-background mouse strain, recapitulating the human CD8-IFN-γ pathway [21]. The strategy consisted of the transient inoculation of B16F10 melanoma cells and depletion of CD4+ regulatory T cells. In detail, after two weeks of acclimatization, vitiligo was induced by inoculation intradermally on the left flank with 2 × 105 B16F10 cells (Day 0), and then CD4-depletion antibodies (BioLegend, San Diego, CA, USA) were injected on Days 4 and 10 to eliminate regulatory T cells. The tumor was surgically excised on Day 12. The turnover rate of skin epidermal cells was approximately 30 days [22]. Epidermal melanocyte loss peaked 30 days after surgery. At that time, the development of vitiligo in MAV mice was monitored and the efficiency of vitiligo induction was then evaluated according to the protocol [15]. Subsequently, mice with vitiligo were assigned to groups by matching the overall level of vitiligo between the groups.

Melanocyte-associated vitiligo (MAV) mice received an intraperitoneal injection of 1,25(OH)₂D₃ at a dose of 5 μg/kg (*n* = 5) or 1 μg/kg (*n* = 5) every day for four weeks or an equivalent volume of 0.9% NaCl (*n* = 5) as a control treatment. Weight- and age-matched female mice receiving placebo treatment were used as the control group for the construction of the vitiligo model. Weight- and age-matched male mice were used as controls for the vitiligo model construct, and these mice did not receive any treatment. On the 30th day of treatment, mice in the above four groups were euthanized to collect blood samples, skin and spleens. The blood sample was collected into a heparinized tube and centrifuged at 3000 rpm for 15 min. The separated plasma was stored frozen at –20 °C for subsequent determination of 1,25(OH)₂D₃ level. The perilesional skin of mice was fixed in 10% formalin for 24 h and embedded in paraffin for histological evaluation. Subsequently, IFN+ CD8+ T cells in the spleen were isolated and analyzed using flow cytometry. Figure 2 details a timeline of vitiligo induction procedures and the assignment of groups.

All the experimental procedures for mice were performed in accordance with the guidelines of the Institutional Animal Care and Use Committee. The protocol was approved by the Committee on the Ethics of Animal Experiments of Huazhong University of Science and Technology (No. 3334).

#### 2.2.3. Cell Viability

Primary CD8+ T cells were inoculated into 96-well culture plates at a density of 2.0 × 10^3^ cells/well, and cell viability was measured using a CCK-8 assay. Treated with different concentrations (1 nM, 10 nM, 100 nM, 1000 nM) of 1,25(OH)₂D₃ (its solvent was used as a control), the CD8+ T cells isolated from vitiligo patients were incubated with CCK-8 (Cell Counting Kit-8, Yeasen, Shanghai, China) solution (20 μL/well) at the specified time points (0, 24, 48, 72, 96, 120 h) and cultured at 37 °C in the dark for 3 h. After finding the appropriate treatment time (72 h) and concentration (10 nM and 100 nM), the CCK-8 assay was re-performed on CD8+ T cells isolated from vitiligo patients and healthy controls. The absorbance was measured at a wavelength of 450 nm using a microplate instrument (BioTek, Winooski, VT, USA). The cell viability rate was calculated as follows: absorbance of drug-treated sample/absorbance of control sample ×100%.

#### 2.2.4. Enzyme-Linked Immunosorbent Assay (ELISA)

The ELISA measurements were conducted in strict accordance with the manual of the experimental kit (Sangon Biotech, Shanghai, China). The optical density (OD) values were measured at a wavelength of 450 nm using a multifunction plate reader (BioTek, Winooski, VT, USA). The standard curve was drawn based on the measured OD values, and the levels of IFN-γ and 1,25(OH)₂D₃ in plasma and culture media supernatants were calculated using the standard curve.

#### 2.2.5. Flow Cytometry Analysis

Cell suspensions were prepared by dicing harvested spleens with a razor blade, digesting with 0.25% Trypsin (NCM Biotech, Suzhou, China) for 30 min at 37 °C, followed by passage through a 40 μM Nylon filter (Thermo Fisher Scientific, Waltham, MA, USA). Mononuclear cells were obtained via density gradient centrifugation with lymphocyte cell separation media (Solarbio, Beijing, China). Cells were then washed in MACS buffer (Miltenyi Biotech, Auburn, CA, USA) after which CD8+ T cells were magnet-based separated using a MojoSort™ Mouse CD8 T Cell Isolation Kit (BioLegend, San Diego, CA, USA). For intracellular cytokine staining, isolated CD8+ T cells were stimulated using a Streptamer^®^ CD3/CD28 kit as previously described and maintained in the presence of 50 IU/mL rIL-2 for 24 h prior to intracellular cytokine staining. Then, single-cell suspensions were stimulated with 50 ng/mL phorbol myristate acetate (PMA) (Millipore, Shanghai, China), 350 ng/mL ionomycin (Thermo Fisher, Wuhan, China) and 800 nl/mL GolgiPlug (BD Biosciences, San Jose, CA, USA) for 4 h. After stimulation, cells were stained with anti-CD8 APC (Biolegend, Beijing, China), fixed and permeabilized, and finally stained with anti-IFN-γ PE (Biolegend, Beijing, China) in accordance with the manufacturer’s instructions. Annexin V-FITC/propidium iodide double-staining was employed to quantify the apoptosis of NHEMs treated with 1,25(OH)₂D₃ and co-cultured with or without CD8+ T cells. The cells were then stained using an Annexin V-FITC/propidium iodide double-fluorescence apoptosis detection kit (BD Biosciences, San Jose, CA, USA) following the manufacturer’s instructions. Flow cytometry was performed using a flow cytometer (BD Biosciences, San Jose, CA, USA) and then analyzed using FlowJo software (version 10.0).

#### 2.2.6. Glucose Uptake Assay

The glucose uptake ability of cells was evaluated using the fluorescent glucose 2-NBDG (Proteintech, Wuhan, China). After different treatments, Tall-104 cells were gently washed with PBS and starved in RPMI-1640 without glucose (Pricella, Wuhan, China) for 2 h. They were subsequently incubated with 2-NBDG (2-NBD-Glucose, 150 μg/mL) for 40 min at 37 °C. Consequently, the cells were rewashed and fluorescence was measured using a Microplate Reader (Bio-Rad, Hercules, CA, USA) at an excitation wavelength of 485 nm and emission wavelength of 535 nm. Fluorescence images were captured using a fluorescence microscope (Nikon, Japan).

#### 2.2.7. Enzyme Activity Assays

The glycolytic capacity of TALL-104 cells was evaluated according to the enzyme activities of lactic dehydrogenase (LDH) and hexokinase (HK) using commercial assay kits (Boxbio, Beijing, China) according to the manufacturer’s instructions. All experiments were normalized using the cell numbers.

#### 2.2.8. Quantitative Real-Time Polymerase Chain Reaction (qRT-PCR)

After treatment with different concentrations of 1,25(OH)₂D₃ and glucose for 72 h, as mentioned above, the cells were harvested for RNA isolation using RNA-easy Isolation Reagent (Vazyme, Nanjing, China) according to the manufacturer’s instructions. RNA concentration and purity were determined by assessing the absorbance at 260 nm and absorbance ratio at 260/280 nm of each sample, respectively (NanoDrop 2000). The samples were reverse-transcribed using an Evo M-MLV reverse transcription Kit II (Accurate Biology, Wuhan, China). Thermal cycling was initiated at 95 °C for 30 s, followed by 40 cycles of 95 °C for 5 s and 60 °C for 30 s. The quantitative real-time PCR (BioRad, San Francisco, CA, USA) was performed using a SYBR Green Pro Taq HS premixed qPCR kit (SAccurate biology, Wuhan, China). Primer sequences were designed by Sagon Bio (Shanghai, China) and are listed as follows in Table 1.

#### 2.2.9. Western Blot Analysis

Tall-104 cells were rinsed twice with PBS. Total protein was extracted using RIPA lysis buffer (Servicebio, Wuhan, China) at 4 °C for 15 min and quantified using a bicinchoninic acid assay (BCA) (Sagon Bio, Shanghai, China). Then, 5× protein loading buffer was added to the lysates prior to their full denaturation in boiling water for 10 min. A total of 25 μg of protein was resolved on 10% sodium dodecyl sulfate-polyacrylamide gel electrophoresis and transferred onto a polyvinylidene fluoride membrane (BD Biosciences, San Jose, CA, USA). The membrane was blocked in 5% bovine serum albumin for 1 h and then incubated at 4 °C overnight with primary antibodies, including AMPK (1:1000, ABclonal, Wuhan, China) and *p*-AMPK (1:1000, ABclonal, Wuhan, China). β-actin (1:2000, ABclonal, Wuhan, China) was used as an internal reference. HRP Goat-anti-Rabbit IgG (1:3000, Servicebio, Wuhan, China), as the secondary antibody, was incubated with the membrane for 90 min at room temperature. Finally, the protein bands were detected using Bio-Rad Chemi-DocXRS (Bio-Rad, Hercules, CA, USA).

#### 2.2.10. Immunohistochemistry

Paraffin slides from vitiligo-model mice (*n* = 3) and healthy control mice (*n* = 3) were submitted to subsequent steps of deparaffinization, rehydration, blocking of endogenous peroxidase activity and incubation in antigen-retrieval solution. After permeabilization and blocking, these sections were incubated with the primary antibodies anti-CD8α mouse mAb (1:100, Servicebio, Wuhan, China) and IFN-γ rabbit pAb (1:100, Abclonal, Wuhan, China) followed by a universal HRP-labeled secondary antibody (1:500, Servicebio, Wuhan, China) and visualized with DAB substrate.

### 2.3. Data Sources and Analysis

#### 2.3.1. Data Source and Bioinformatic Analysis

The expression profiles GSE94138 were obtained from the Gene Expression Omnibus (https://www.ncbi.nlm.nih.gov/geo/query/acc.cgi?acc=GSE94138 (accessed on 1 February 2022)), and the matrix included transcription profile data of PBMCs from 47 participants receiving a weekly dose of 20,000 IU of 1,25(OH)₂D₃ and 47 controls receiving placebo for three to five years. R software (version 4) and bioconductor packages were used in data mining, excavating and statistical analyses. The limma package was subsequently used for the calculation of aberrantly expressed mRNAs and to identify DEGs between the 1,25(OH)₂D₃ supplement group and controls with the cut-off criterion of *p* < 0.05 and absolute log2FC > 2. The volcano plots and heatmap were generated using the ggplot2 package. Kyoto Encyclopedia of Genes and Genomes (KEGG) enrichment analyses of DEGs were conducted utilizing the cluster Profiler R package.

#### 2.3.2. Data Presentation and Statistical Analysis

Statistics are displayed as mean ± SD for continuous variables and N (%) for categorical variables. Statistical analyses and graphical visualizations were performed using GraphPad Prism (v.8.1.0). For continuous variables, comparisons were made using parametric paired *t*-tests or non-parametric Mann–Whitney tests, as appropriate. Statistical tests for categorical variables were performed using the Chi-square test. A non-parametric Spearman correlation test was used to analyze the bivariate correlations. *p*-values < 0.05 were considered statistically significant. Sample size calculation was determined using PASS 15.0 software. For the representative images and FACS profiles presented in the paper, each experiment was repeated at least three times independently with similar results.

## 3. Results

### 3.1. Deficiency of 1,25(OH)₂D₃ Is Associated with Disease Activity and Dysfunction of CD8+ T Cells in the Vitiligo Cohort

In this cross-sectional study, we collected clinical data from 327 patients with vitiligo as illustrated in Table 2. The vitiligo cohort with a mean serum 1,25(OH)₂D₃ concentration of 16.73 ± 6.353 (mean ± SD) was further divided into a 1,25(OH)₂D₃-deficient (<20 ng/mL) group and a 1,25(OH)₂D₃-sufficient (≥20 ng/mL) group. In vitiligo patients, the percentage of 1,25(OH)₂D₃ deficiency was significantly higher compared to that in the 1,25(OH)₂D₃-sufficient group (71.62% vs. 28.74%). Serum 1,25(OH)₂D₃ levels in these two groups showed non-significant differences concerning age, disease duration, gender, insomnia, alcohol consumption, smoking, family history, VASI and clinical type of disease. However, VIDA scores were significantly higher in the 1,25(OH)₂D₃-deficient group (mean ± SD, 2.991 ± 1.196) than in the 1,25(OH)₂D₃-sufficient group (mean ± SD, 2.853 ± 1.247, *p* = 0.0024). Furthermore, by collecting and observing the immunological characteristics of the patients, although there was no significant difference in total counts of T cells between the two groups (mean ± SD, 1372 ± 527.5 and 1470 ± 503.7, *p* =0.0830), a significantly higher frequency of CD8+ T cells and IFN + CD8+ T cells were found in the 1,25(OH)₂D₃-deficient group (mean ± SD, 24.96 ± 5.834% and 39.67 ± 11.61%) versus the 1,25(OH)₂D₃-sufficient group (mean ± SD, 22.88 ± 5.444% and 36.26 ±1 5.46%, *p* = 0.0019 and *p* = 0.0011). To further investigate the relationship between levels of serum 1,25(OH)₂D₃ and immunological characteristics in vitiligo, the correlation between serum 1,25(OH)₂D₃ levels and the three immunological profiles mentioned above were analyzed, and the results shown in Figure 3a–c revealed a significantly negative correlation between 1,25(OH)₂D₃ and counts of T cells (r = −0.1179, *p* = 0.0012) and frequency of CD8+ T (r = −0.1734, *p* = 0.0016) and IFN + CD8+ T cells (r = −0.1895, *p* = 0.0006).

### 3.2. 1,25(OH)₂D₃ Influences Glucose Metabolism and Signaling Pathways of Circulating Immune Cells

By excavating the public database, the expression profiles GSE94138, which enrolled participants receiving vitamin D (*n* = 47) or placebo (*n* = 47), were found. Differentially expressed genes (DEGs) were analyzed in the sequencing data of the PBMC samples in the dataset and 244 aberrantly expressed genes were identified, including 117 upregulated and 127 downregulated genes (Appendix A). Figure 3d presents the volcano plot built for all identified genes showing the relative increase (red) or decrease (blue) compared with the placebo group (log2FC > 2, *p* < 0.05). To further understand the gene differences between the two groups, the top 50 most significantly changed genes were visualized in a clustering heatmap, which directly revealed the variation in each gene (Figure 3e). Kyoto Encyclopedia of Genes and Genomes (KEGG) pathway enrichment analysis indicated that 1,25(OH)₂D₃ supplementation exhibited changes by influencing multiple signaling pathways, which included the glycolysis/gluconeogenesis and AMPK signaling pathways (Figure 3f).

### 3.3. 1,25(OH)₂D₃ Inhibits Proliferation and Cytotoxicity of CD8+ T Cells

The effects of different concentrations of 1,25(OH)₂D₃ on CD8+ T cells at different time points were detected using a CCK-8 assay. The maximum inhibition of 1,25(OH)₂D₃ on CD8+ T cell proliferation was observed when treated for 72 h. The inhibition of CD8+ T cell proliferation increased with increasing concentrations of 1,25(OH)₂D₃ between 1 nM and 100 nM. However, when the concentration of 1,25(OH)₂D₃ was 1000 nM, the inhibitory effect on cell proliferation was attenuated compared to concentrations below 1000 nM (Figure 4a). We further investigated the inhibitory effect of 1,25(OH)₂D₃ on IFN-γ secretion after treating CD8+ T cells for 72 h and found that there was a strong dose-dependent relationship between the concentration of 1,25(OH)₂D₃ and the IFN-γ secretion ability of CD8+ T cells. Similar to the inhibition of proliferation, the IFN-γ secretion capacity of CD8+ T cells was also significantly inhibited at 10 nM and 100 nM concentrations of 1,25(OH)₂D₃. However, at a concentration of 1000 nM, the inhibitory effect of 1,25(OH)₂D₃ on IFN-γ secretion was alleviated (Figure 4b). Based on the above results, we further verified that CD8+ T cells were the target cells for 1,25(OH)₂D₃-induced immunosuppressive activity in vitiligo by applying 1,25(OH)₂D₃ at concentrations of 10 nM and 100 nM to CD8+ T cells isolated from vitiligo patients and healthy controls for 72 h. Here, we observed that CD8+ T cells in the vitiligo group were significantly activated, with higher proliferation rates (*p* = 0.0351) and more IFN-γ production (*p* = 0.0190). The inhibitory effect of 1,25(OH)₂D₃ at a concentration of 100 nM was more effective in inhibiting CD8+ T cell proliferation (*p* = 0.0227) and IFN-γ production (*p* = 0.0216) than 1,25(OH)₂D₃ at a concentration of 10 nM (Figure 4c,d). To determine the effect of 1,25(OH)₂D₃ on the role of CD8+ T cells leading to the apoptosis of NHEMs, we examined the apoptosis rate of NHEMs in the presence of 10 nM or 100 nM 1,25(OH)₂D₃ with or without CD8+ T cells. As shown in Figure 4e, CD8+ T cells from patients with vitiligo resulted in a significant upregulation of the apoptosis rate of NHMES cells after excluding the effect of 10 nM (*p* = 0.0138) or 100 nM 1,25(OH)₂D₃ (*p* = 0.0326). The apoptotic effect of CD8+ T cells on NHMES cells was strongly reversed by the addition of 100 nM 1,25(OH)₂D₃ (*p* = 0.0046). However, 10 nM 1,25(OH)₂D₃ was unable to reverse the apoptotic effect (*p* = 0.3071).

### 3.4. 1,25(OH)₂D₃ Inhibits Glycolysis of CD8+ T Cells by Regulating the AMPK Pathway

First, we found that glucose concentrations of 5 mM and 10 mM did not significantly affect the proliferation of CD8+ T cells in the pre-experiment (Appendix A), so we chose 5 mM as the low-glucose control group and 10 mM as the normal-glucose-concentration group to serve as the basic glucose concentration for detecting the glucose metabolism of CD8+ T cells. The human MHC non-restricted cytotoxic CD8+ T cell leukemia line TALL-104 cells were cultured with or without the presence of 10 nM or 100 nM 1,25(OH)₂D₃ in normal 10 mM glucose-containing medium, and medium without 1,25(OH)₂D₃ and containing 5 mM glucose was used as a negative control. The fluorescent glucose 2-NBDG was used to measure the glucose uptake in TALL-104 cells. After treatment with 1,25(OH)₂D₃ for 72 h, compared with the control group, the glucose uptake of TALL-104 cells in the 5 nM glucose group (*p* = 0.0078) and the 100 nM 1,25(OH)₂D₃ group (*p* = 0.0008) was significantly decreased. Furthermore, the decrease was more significant in the 100 nM 1,25(OH)₂D₃ group than in the 10 nM 1,25(OH)₂D₃ group (*p* = 0.0324, Figure 5a,b). The activities of the glycolytic enzymes lactate dehydrogenase (LDH) and hexokinase (HK) exhibited parallel changes as described above, indicating that 1,25(OH)₂D₃ decreased the glycolysis level in TALL-104 cells (Figure 5c,d). Moreover, 1,25(OH)₂D₃ decreased the level of IFN-γ secreted by TALL-104 cells in a dose-dependent manner after 72 h of culture (Figure 5e). Interestingly, in terms of gene expression, the mRNA levels of GLUT1 (*p* = 0.0380), LDHA (*p* = 0.0058) and IFN-γ (*p* = 0.00167) were significantly downregulated after 72 h of 100 nM 1,25(OH)₂D₃ treatment compared to 10 nM 1,25(OH)₂D₃ treatment, whereas no significant differences were observed in the mRNA expression levels of HKII (*p* = 0.2894, Figure 5f). To investigate the essential signaling pathways involved in the effect of 1,25(OH)₂D₃ on glycolysis, the protein phosphorylation of AMPK was evaluated, which was discovered via KEGG pathway enrichment by excavation of the public database. Upregulated *p*-AMPK/AMPK ratios in TALL-104 cells were observed at 1,25(OH)₂D₃ concentrations of both 10 nM (*p* = 0.0194) and 100 nM (*p* = 0.0148, Figure 5g). Tall-104 cells were then treated with the *p*-AMPK inhibitor Compound C to determine the role of AMPK in glucose metabolism after 1,25(OH)₂D₃ treatment. Western blot analysis revealed that both basal phosphorylation levels (*p* = 0.0104) and increased phosphorylation levels induced by 1,25(OH)₂D₃ (*p* = 0.0132) were reduced when Compound C was added (Figure 5h). In agreement with the Western blot analysis, measurements of glucose uptake (*p* = 0.0010), LDH activity (*p* = 0.0022), HK activity (*p* = 0.0018) and IFN-γ secretion (*p* = 0.0213) in Tall-104 cells confirmed that downregulated glycolysis and cytotoxicity via 1,25(OH)₂D₃ could be reversed by inhibiting phosphorylation levels of AMPK with Compound C (Figure 5i).

### 3.5. Therapeutic Effect of 1,25(OH)₂D₃ on Melanocyte-Associated Vitiligo Mice

MAV mice were constructed based on endogenous autoreactive CD8+ T cells targeting skin melanocytes. After treatment with 1,25(OH)₂D₃ for one month, we observed remarkable repigmentation, especially in the 5 μg/kg 1,25(OH)₂D₃-treated group, as shown in Figure 6a. Serum 1,25(OH)₂D₃ levels were measured after treatment (Figure 6b). There were significant differences in serum 1,25(OH)₂D₃ concentrations between MAV mice and the control group (*p* = 0.0045) and the two 1,25(OH)₂D₃ treatment groups (*p* = 0.0004). Immunohistochemistry revealed 1,25(OH)₂D₃ treatment reduced CD8+ T cell infiltration in the vitiligo mouse model. Specifically, both 1,25(OH)₂D₃ treatment groups exhibited fewer CD8+ T cells in their lesions than the MAV-mice group, and fewer CD8+ T cells were found in the 5 μg/kg 1,25(OH)₂D₃-treated group compared with the 1 μg/kg 1,25(OH)₂D₃-treated group (*p* = 0.0037, Figure 6c). In vitiligo perilesional skin, IFN-γ levels were significantly decreased in the skin of 5 μg/kg 1,25(OH)₂D₃-treated mice (*p* = 0.0019), which was more pronounced than that in the skin of mice treated with 1 μg/kg 1,25(OH)₂D₃ (*p* = 0.0193, Figure 6d). By sorting CD8+ T cells from the spleen of MAV mice or controls, the frequency of the IFN-γ + CD8+ T cell population was found to be decreased in both treatment groups, with a more significant decrease observed in the 5 μg/kg 1,25(OH)₂D₃-treated group than in the 1 μg/kg 1,25(OH)₂D₃-treated group (*p* = 0.0005, Figure 6e).

## 4. Discussion

Recently, strong clinical associations between 1,25(OH)₂D₃ status and the incidence/severity of many autoimmune disorders have prompted the idea of utilizing 1,25(OH)₂D₃ supplementation to manipulate disease outcomes [23]. While much is known about the effects of 1,25(OH)₂D₃ on innate immune responses and helper T cell immunity, relatively limited progress has been made on the frontier of cytotoxic T lymphocyte immunity—an effector arm of host cellular adaptive immunity that is crucial to the pathogenesis of numerous autoimmune diseases [24].

Accumulating evidence shows that autoreactive cytotoxic CD8+ T cells promote disease progression via the local production of IFN-γ in vitiligo. Both topical and systemic treatments that block IFN-γ signaling produced by CD8+ T cells can effectively reverse depigmentation, such as JAK inhibitors [25]. Our study demonstrates that serum 1,25(OH)₂D₃ levels correlate with IFN-γ+ CD8+ T cell activity in vitiligo and may serve as an important biomarker of vitiligo severity. The modulation of autoreactive CD8+ T cell proliferation and IFN-γ secretion by 1,25(OH)₂D₃ may be a novel approach for the treatment of vitiligo.

The latest studies do not support a causal role of vitamin D levels in vitiligo, which contradicts previous epidemiologic reports that vitiligo patients are associated with vitamin D deficiency [10]. Therefore, we tested the relationship between the level of serum vitamin D and vitiligo in a cross-sectional study and found that, compared to patients with sufficient 1,25(OH)₂D₃, the 1,25(OH)₂D₃-deficient group had significantly higher VIDA scores and increased percentages of CD8+ T cells/T cells and IFN−γ+ CD8+ T cells/CD8+ T cells in the blood, indicating specific immune abnormalities. Meanwhile, we found a significant negative correlation between the serum level of 1,25(OH)₂D₃ and the above immune profiles in vitiligo patients. Studies have shown that 1,25(OH)₂D₃ plays an immunomodulatory role via the regulation of T cell function and inhibition of T cell proliferation and cytokine production [26]. The number of CD8+ T cells is also higher in patients with vitiligo compared to healthy individuals, and in addition, it correlates with the severity of the disease. Research found that, in patients in the progressive state, there were significantly more CD8+ cytotoxic T cells, which also expressed significantly higher levels of IFN-γ [11]. Given our findings in the clinical study, we suppose that 1,25(OH)₂D₃ deficiency may be an important factor contributing to the initiation and progression of vitiligo, as 1,25(OH)₂D₃ may inhibit CD8+ cytotoxic T cell hyperactivation.

Here, we have identified that 1,25(OH)₂D₃ can reduce the cytotoxicity of activated alloreactive T cells, as determined by decreased proliferation and IFN-γ secretion. We further investigated the potential regulation mechanism of 1,25(OH)₂D₃ via excavating the public database and thus found that the DEGs were enriched in glycolysis. For effector CD8+ T cells, changes in their glycolysis play an important role in the production of IFN-γ, whereas downregulation of glycolysis levels is detrimental to their production of relevant cytokines and immune functions [27]. The remodeling of glucose metabolism in pathogenic CD8+ T cells can reduce the deleterious inflammatory and cytolytic effects in autoimmune diseases such as rheumatoid arthritis [28]. Thus, the promotion of a metabolic shift to reduce glycolytic activity may protect patients from disease progression, and the modulation of glucose metabolism could be a potential therapeutic target for T cell-mediated disease.

In the study of vitiligo, little research has focused on the glucose metabolism of CD8+ T cells. The TALL-104 cell line was used in this study as a cell model for cytotoxic CD8+ T lymphocytes; it is a CD3+, CD8+, TCRα/β+ cytotoxic T cell that proliferates infinitely in vitro without additional specific activation and secretes a large number of cytokines, such as IFN-γ [29,30]. We found that 1,25(OH)₂D₃ treatments downregulated glucose uptake and the expression of the glycolytic enzymes of TALL-104 cells, suggesting that 1,25(OH)₂D₃ may suppress glycolysis. Glucose transported into the cell via GLUT is phosphorylated by HKs, the first rate-limiting enzyme in the glycolytic pathway and critical for glycolysis. LDH is also one of the crucial enzymes in aerobic glycolysis, catalyzing the last step of glycolysis. The activity and mRNA expression levels of GLUT1, HK-II and LDHA were significantly decreased in TALL-104 cells after 1,25(OH)₂D₃ treatment in a dose-dependent manner. The AMPK pathway, a master sensor of cellular energy states, was also significantly enriched in the aforementioned data excavation. The activation of AMPK can restore energy balance by inhibiting anabolism and enhancing catabolism to produce energy, interfering with T cell activation and proliferation and subsequently inhibiting glycolysis [31]. A recent study showed that AMPK is required for metabolic adaptation in activated T cells. The dysregulation of energy homeostasis in CD8+ T cells is thought to be an important factor in driving changes in autoimmune diseases such as type 2 diabetes [32]. The role of the AMPK signaling pathway has made it an attractive target for drugs aimed at preventing and/or treating autoimmune diseases [33]. We hypothesize that 1,25(OH)₂D₃ inhibits glycolysis by activating the AMPK signaling pathway. The role of AMPK signaling was verified using Western blot analysis, and it may be an important mechanism for the inhibition of glycolysis and the immune function of CD8+ T cells by 1,25(OH)₂D₃. Compound C is the only available agent that is used as a cell-permeable AMPK inhibitor [34]. Using functional rescue experiments with Compound C pretreatment, we demonstrated that promotion of AMPK phosphorylation is critical for 1,25(OH)₂D₃ to produce the above effects. We disclose that the AMPK pathway may be the exact mechanism by which 1,25(OH)₂D₃ inhibits CD8+ T cell cytotoxicity in vitiligo patients via regulation of glucose metabolism.

Because of the complex pathogenesis of vitiligo, the management of it remains challenging. The treatment of vitiligo remains challenging due to its complex pathogenesis. Studies have found that vitamin D supplementation has therapeutic effects in different experimental animal models, such as allergic encephalomyelitis, collagen-induced arthritis, type 1 diabetes, inflammatory bowel disease, autoimmune thyroiditis and systemic lupus erythematosus [35]. Therefore, vitamin D supplementation has the potential to be used in the treatment of autoimmune diseases such as vitiligo. Translational research using human samples often lacks insight into mechanisms, which basic research is better equipped to address. The development and use of animal models of vitiligo have helped to increase the understanding of mechanisms and treatments based on clinical observations. In this study, we clarified that 1,25(OH)₂D₃ ameliorated ongoing depigmentation and decreased the infiltrating CD8+ T cell numbers in the skin by using the MAV mouse model. In this model, a large number of autoreactive CD8+ T cells are locally recruited in the skin and produce IFN-γ to drive patterned skin autoimmunity, which is similar to the initiation and progression of human vitiligo [15]. We demonstrated that 1,25(OH)₂D₃ could inhibit the proliferation and recruitment of CD8+ T cells in vivo. In addition, it abrogates the capacity of CD8+ T cells to secrete IFN-γ upon activation. These effects indicate that 1,25(OH)₂D₃ has the potential for clinical application in managing vitiligo and may be a potential therapeutic target for other autoimmune diseases mediated by CD8+ T cells.

The main function of 1,25(OH)₂D₃ is maintaining the proper levels of calcium and phosphorus in serum. Thus, higher doses of 1,25(OH)₂D₃ can cause hypercalcemia as a side effect [36]. However, a prospective study validated the therapeutic efficacy of oral vitamin D in pediatric vitiligo patients and found that vitamin D supplementation at doses higher than the standard dose was not sufficient to cause hypercalcemia, which may be more useful in the treatment of pediatric vitiligo patients with concomitant vitamin D deficiency [37]. Another study enrolling 16 patients with vitiligo who received 35,000 IU 1,25(OH)₂D₃ once daily for six months found no change in serum calcium (total and ionized) and an increase in urinary calcium excretion within the normal range of patients, suggesting that high-dose vitamin D_3_ therapy may be effective and safe for vitiligo [38]. Analogs of 1,25(OH)₂D₃ are currently being developed to target specific diseases with minimal side effects [39]. More robust clinical trials with a large number of subjects are needed to confirm a safe dose range of 1,25(OH)₂D₃. Moreover, the concrete mechanism underlying the modulation of CD8+ T cell glucose metabolism by 1,25(OH)₂D₃ in vitiligo has not been further explored in this study. The therapeutic values of 1,25(OH)₂D₃ for vitiligo should also be replicated in larger populations.

## 5. Conclusions

Overall, we demonstrate that 1,25(OH)₂D₃ is effective in treating vitiligo by mediating the inhibition of glycolysis and the effector function of CD8+ T cells via AMPK signaling (Figure 7). Our results potentially provide a partial explanation for the underlying mechanisms controlling the effectiveness of 1,25(OH)₂D₃ and provide evidence that 1,25(OH)₂D₃ may be translated for its pharmacological activity against vitiligo, which holds promise for the future treatment of various autoimmune skin conditions.

## Figures and Tables

**Figure 1 nutrients-15-04697-f001:**
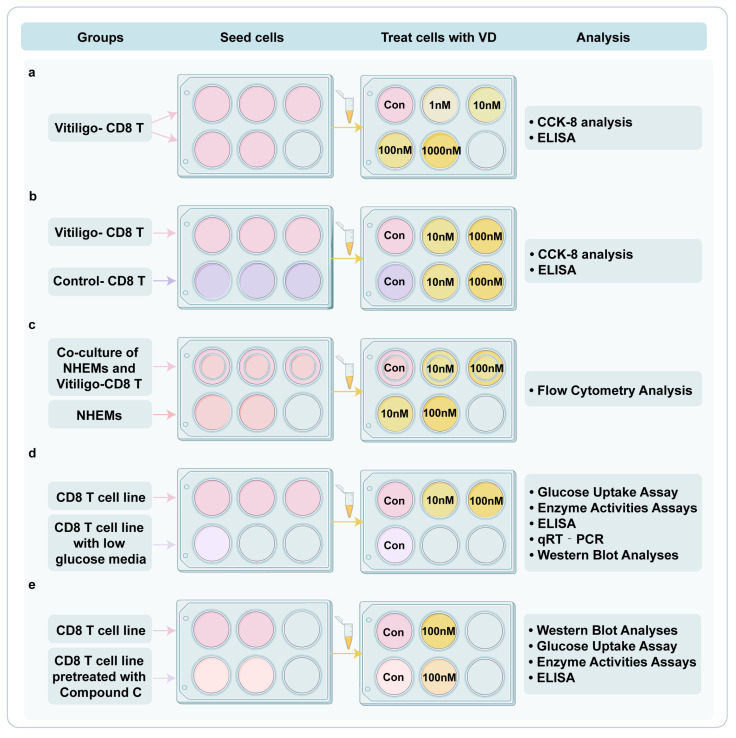
In vitro experiments. (**a**). CD8+ T cells were isolated from vitiligo patients and treated with different concentrations (1 nM, 10 nM, 100 nM, 1000 nM) of 1,25(OH)₂D₃ and its solvent as control. Then, the cell viability was detected using a CCK-8 assay, and the level of IFN-γ in the supernatant was detected using ELISA. (**b**). CD8+ T cells were isolated from vitiligo patients and healthy controls and treated with 10 nM and 100 nM 1,25(OH)₂D₃ and its solvent as control, respectively, and then the cell viability was detected using a CCK-8 assay, and the level of IFN-γ in the supernatant was detected using ELISA. (**c**). The apoptosis of NHEMs treated with 10 nM and 100 nM concentrations of 1,25(OH)₂D₃, whether co-cultured with CD8+ T cells or not, was detected using a flow-through assay (NHEMs co-cultured with CD8+ T cells without 1,25(OH)₂D₃ treatment as control). (**d**). The CD8+ T cell line was treated with 10 nM and 100 nM 1,25(OH)₂D₃, using no 1,25(OH)₂D₃ treatment as the negative control and low-glucose medium as the positive control, respectively. Then, the glucose uptake assay was performed and glycolytic enzyme activity was measured, the level of IFN-γ in the supernatant was determined using ELISA, the expression level of mRNA was determined using qRT-PCR, and the protein expression level was determined using Western blotting analysis. (**e**). The CD8+ T cell lines were pretreated with 10 um Compound C or not and then treated with 100 nM 1,25(OH)₂D₃, respectively. All in vitro experiments had at least three replicates in each group. NHEMs, normal human epidermal melanocytes; VD, 1,25(OH)_2_D_3_; ELISA, enzyme-linked immunosorbent assay; qRT-PCR, quantitative real-time polymerase chain reaction.

**Figure 2 nutrients-15-04697-f002:**
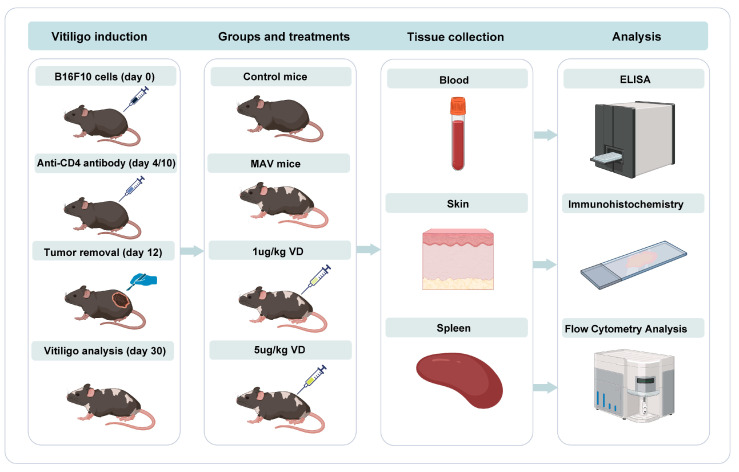
Animal model experiments. For animal experiments, vitiligo was induced by inoculating B16F10 cells (Day 0), and then CD4-depletion antibodies were injected on Days 4 and 10. The tumor was surgically excised on Day 12. MAV mice were assigned to groups by matching the overall level of vitiligo between the groups and received an intraperitoneal injection of 1,25(OH)₂D₃ at a dose of 5 μg/kg (*n* = 5) or 1 μg/kg (*n* = 5) every day for four weeks or an equivalent volume of 0.9% NaCl (*n* = 5) as a control treatment. Weight- and age-matched female mice receiving placebo treatment were used as the control group for construction of the vitiligo model. On the 30th day of treatment, mice in the above four groups were euthanized to collect blood samples, skin and spleens. The separated plasma was stored frozen at −20 °C for subsequent determination of 1,25(OH)₂D₃ level. Perilesional skin of mice was used for histological evaluation. Subsequently, IFN+ CD8+ T cells in spleens were isolated and analyzed using flow cytometry. MAV, melanocyte-associated vitiligo; VD, 1,25(OH)_2_D_3_; ELISA, enzyme-linked immunosorbent assay; qRT-PCR, quantitative real-time polymerase chain reaction.

**Figure 3 nutrients-15-04697-f003:**
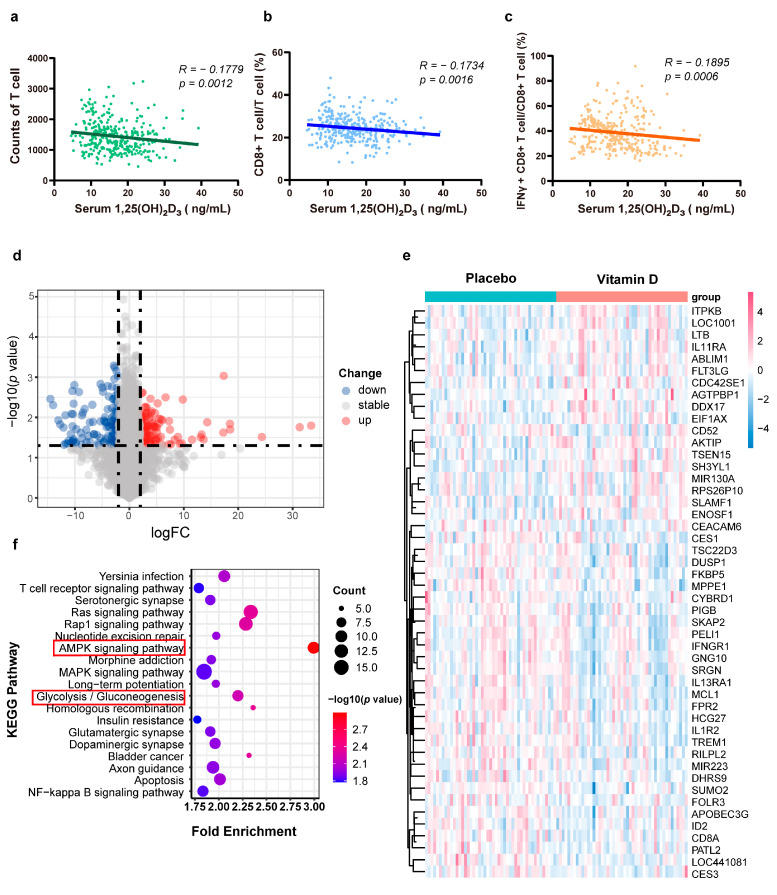
Correlation between 1,25(OH)₂D₃ and immune cells. (**a**–**c**). The relationship between serum 1,25(OH)₂D₃ levels in patients with different immunological data was evaluated and revealed a significant negative correlation between serum 1,25(OH)₂D₃ and the counts of T cells and frequency of CD8+ T cells/ T cells and IFN+ CD8+ T cells/ CD8+ T cells. (**d**). Volcano plot of detectable mRNA profiles in participants receiving vitamin D (*n* = 47) or placebo (*n* = 47). Red and blue plots represent aberrantly expressed mRNAs with *p* < 0.05 and absolute log2FC > 2, and grey plots indicate transcripts not significantly changed. The abscissa means the value of fold change in mRNA expression. The ordinate shows the −log10 of the *p*-value for each mRNA, representing the strength of the association. (**e**). A plot heatmap showing the top 50 gene expression profiles of differentially expressed genes (DEGs). The magnitude of change for each variable is indicated by the color gradient. The most intense shades of red in the red gradient represent the greatest magnitudes of change for the dependent variable. As the color gradient becomes whiter, it indicates no change in the dependent variable. As the color gradient flows from white to more intense shades of blue, it indicates a decrease in the values of the dependent variable. (**f**) The KEGG pathway enrichment analysis revealed that DEGs involved in the glycolysis/gluconeogenesis and AMPK signaling pathways were enriched significantly in the participants receiving vitamin D. AMPK, AMP-activated protein kinase.

**Figure 4 nutrients-15-04697-f004:**
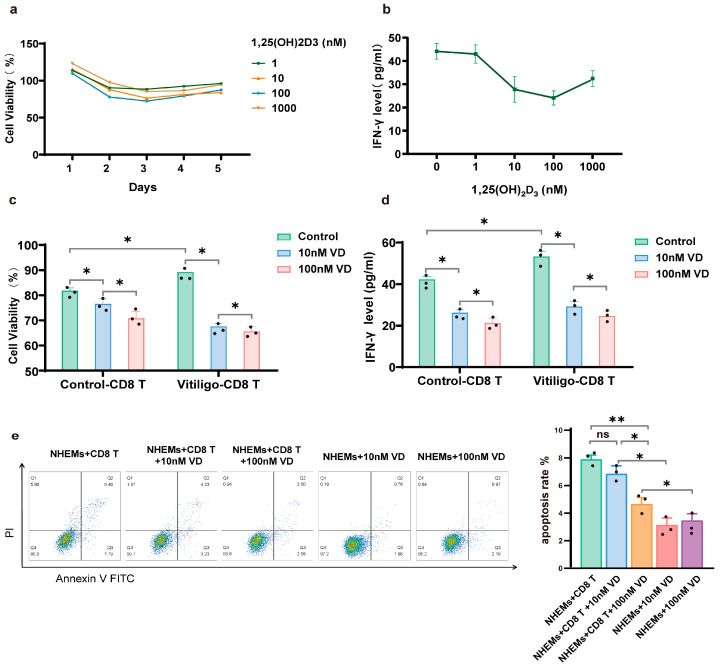
Inhibition of CD8+ T cell function by 1,25(OH)₂D₃. (**a**). Cell viability of CD8+ T cells at different time points at each 1,25(OH)₂D₃ concentration. (**b**). Levels of IFN-γ in CD8+ T cell supernatants after 72 h of 1,25(OH)₂D₃ treatment at different concentrations. (**c**). Cell viability of CD8+ T cells derived from patients with vitiligo and controls after 72 h of 1,25(OH)₂D₃ treatment at different concentrations. (**d**). Levels of IFN-γ in supernatants of CD8+ T cells derived from patients with vitiligo and controls after 72 h of 1,25(OH)₂D₃ treatment at different concentrations. (**e**). NHEM apoptosis was quantified by double-staining with FITC-annexin V and PI immediately after 1,25(OH)₂D₃ treatment at different concentrations with or without a 72 h co-culture period with CD8+ T cells. The data are from a representative experiment out of three performed. The data are presented as the mean ± standard deviation (SD) and individual data points are represented as black dots. *, *p* < 0.05; **, *p* < 0.01; ns, not significant. Annexin V-FITC, Annexin V-fluorescein isothiocyanate; PI, propidium iodide; NHEMs, normal human epidermal melanocytes; VD, 1,25(OH)₂D₃.

**Figure 5 nutrients-15-04697-f005:**
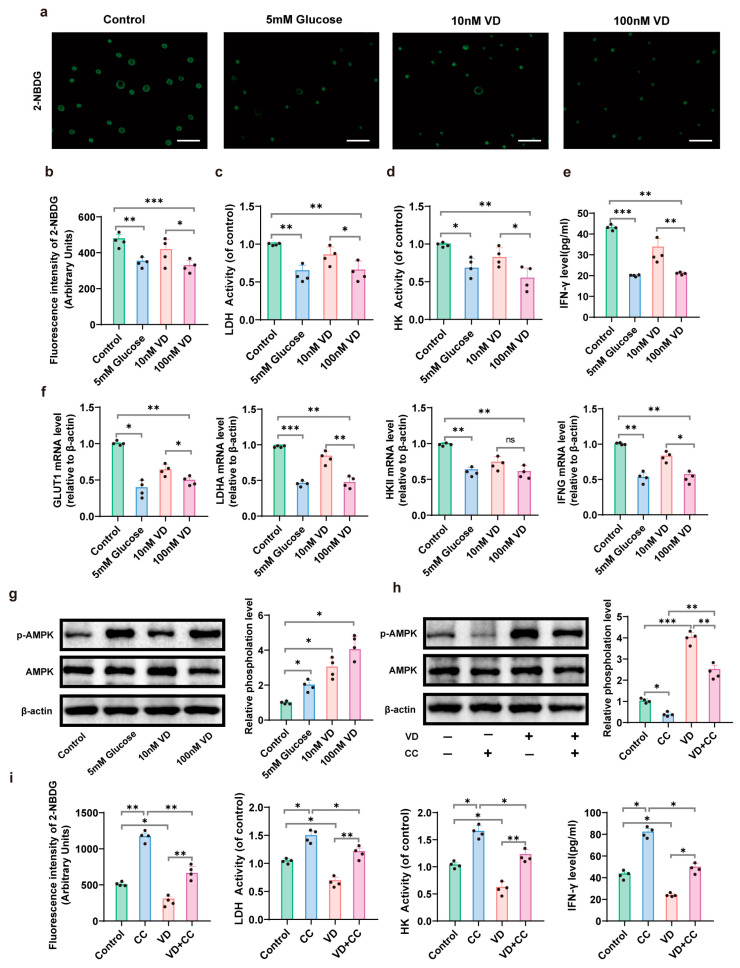
Regulation of CD8+ T cell glucose metabolism and immunity by 1,25(OH)₂D₃ via AMPK signaling pathway. (**a**). The uptake of 2-NBDG by TALL-104 cells in each group was observed using fluorescence assay. (**b**). Fluorescence intensity of 2-NBDG uptake by TALL-104 cells in each group. (**c**). LDH activity of TALL-104 cells in each group. (**d**). HK activity of TALL-104 cells in each group. (**e**). Levels of IFN-γ in the supernatant of TALL-104 cells in each group. (**f**). GLUT1, LDHA, HKII and IFNG mRNA expression in each group. (**g**). The expression and phosphorylation of AMPK in each group. (**h**). The expression and phosphorylation of AMPK were analyzed using Western blotting after treatment of TALL-104 cells with 1,25(OH)₂D₃ and/or Compound C. (**i**). 2-NBDG fluorescence intensity, LDH activity, HK activity and IFN-γ level in supernatant of TALL-104 cells treated with 1,25(OH)₂D₃ and/or Compound C. Data are from a representative one of three experiments and are expressed as mean ± standard deviation (SD). Individual data points are represented as black dots. *, *p* < 0.05; **, *p* < 0.01; ***, *p* < 0.001; 2-NBDG, 2-NBD-Glucose; LDH, lactic dehydrogenase; HK, hexokinase; AMPK, AMP-activated protein kinase; CC, Compound C; VD, 1,25(OH)₂D₃.

**Figure 6 nutrients-15-04697-f006:**
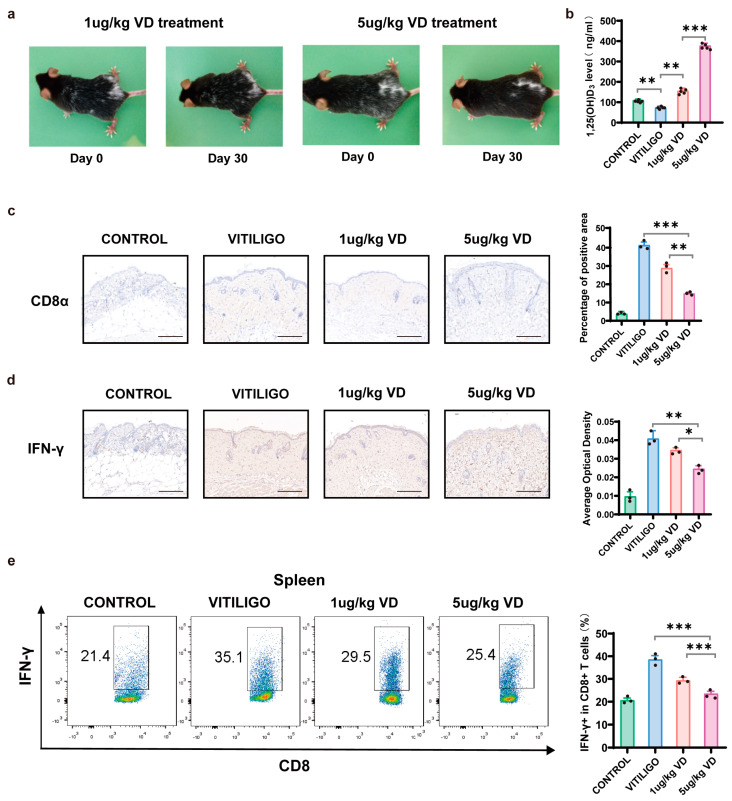
1,25(OH)₂D₃ ameliorated vitiligo lesions in mice and inhibited the function of CD8+ T cells. (**a**). Skin lesions of melanocyte-associated vitiligo (MAV) mice were monitored after the 1-month treatment period with 1,25(OH)₂D₃. Sample photos of MAV mice are shown. (**b**). Comparison of plasma levels of 1,25(OH)₂D₃ between different groups (*n* = 5). (**c**,**d**). Immunohistochemical staining of CD8α and IFN-γ in skin lesions of MAV mice; the percentage of the positive area was measured in each of the four groups (*n* = 3). Representative images are shown; Bar = 200 μm. (**e**). Sample flow plots and quantification of CD8+ T cells producing IFN-γ in the spleen. The data are presented as the mean ± standard deviation (SD) and individual data points are represented as black dots. * *p* < 0.05, ** *p* < 0.01, *** *p* < 0.001. VD, 1,25(OH)₂D₃.

**Figure 7 nutrients-15-04697-f007:**
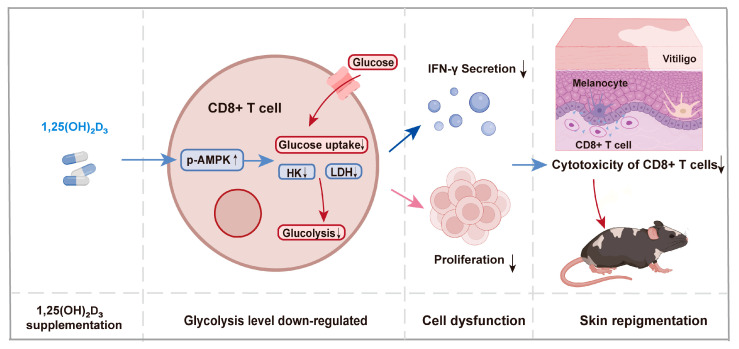
1,25(OH)₂D₃ supplementation can lead to CD8+ T cell dysfunction, resulting in skin repigmentation. Via a supply of exogenous 1,25(OH)₂D₃, the phosphorylation level of AMPK in CD8+ T cells is upregulated, which leads to reduced levels of glycolysis, as evidenced by the downregulation of glucose uptake and activities of the key enzymes of glycolysis, HK and LDH. These changes altered the effect of CD8+ T cells, leading to a decrease in proliferation and IFN-γ secretion of CD8+ T cells, consequently decreasing their cytotoxicity against melanocytes. As described above, the immunomodulatory effect of 1,25(OH)₂D₃ on CD8+ T cells causes skin repigmentation, which we have verified in the MAV mouse model. LDH, lactic dehydrogenase; HK, hexokinase; AMPK, AMP-activated protein kinase.

**Table 1 nutrients-15-04697-t001:** Primer sequences.

mRNA	Forward Primer (5′–3′)	Reverse Primer (5′–3′)
GLUT1	GGCCAAGAGTGTGCTAAAGAA	ACAGCGTTGATGCCAGACAG
LDHA	ATGGCAACTCTAAAGGATCAGC	CCAACCCCAACAACTGTAATCT
HKII	GAGCCACCACTCACCCTACT	CCAGGCATTCGGCAATGTG
IFNG	TCGGTAACTGACTTGAATGTCCA	TCGCTTCCCTGTTTTAGCTGC
β-actin	CATGTACGTTGCTATCCAGGC	CTCCTTAATGTCACGCACGAT

**Table 2 nutrients-15-04697-t002:** Baseline demographics and disease characteristics in 1,25(OH)₂D₃ sufficient and deficient cohorts.

	All (N = 327)	1,25(OH)₂D₃ Sufficient (≥20 ng/mL, N = 94, 28.74%)	1,25(OH)₂D₃ Deficient (<20 ng/mL, N = 233, 71.62%)	*p*
Serum 1,25(OH)₂D₃ concentration, mean ± SD, μg/mL	16.73 ± 6.353	24.82 ± 3.745	13.47 ± 3.730	<0.0001
Disease duration, mean ± SD, month	46.2 6 ± 82.62	50.36 ± 97.79	44.61 ± 75.82	0.1826
total counts of T cells, mean ± SD	1442 ± 511.8	1372 ± 527.5	1470 ± 503.7	0.0830
CD8+ T cells/T cells, mean ± SD, %	24.36 ± 5.793	22.88 ± 5.444	24.96 ± 5.834	0.0019
IFN−γ+ CD8+ T cells/CD8+ T cells, mean ± SD, %	38.69 ± 12.90	36.26 ± 15.466	39.67 ± 11.61	0.0011
VASI, mean ± SD	1.904 ± 5.437	1.422 ± 2.238	2.099 ± 6.277	0.2683
VIDA, No (%)				0.0024
−1	7 (2.14%)	3 (3.19%)	4 (1.72%)	
0	13 (3.98%)	7 (7.45%)	6 (2.58%)	
1	21 (6.42%)	4 (4.26%)	17 (7.3%)	
2	70 (21.41%)	30 (31.91%)	40 (17.17%)	
3	85 (25.99%)	25 (26.6%)	60 (25.75%)	
4	131 (40.06%)	25 (26.6%)	106 (45.49%)	
Types of vitiligo, No (%)				0.7556
Nonsegmental	194 (59.33%)	54 (57.45%)	140 (60.09%)	
Segmental	95 (29.05%)	30 (31.91%)	65 (27.9%)	
Unclassified	38 (11.62%)	10 (10.64%)	28 (12.02%)	
Sex, No (%)				0.0930
Male	157 (48.01%)	52 (55.32%)	105 (45.06%)	
Female	170 (51.99%)	42 (44.68%)	128 (54.94%)	
Smoking, No. (%)				0.2121
No	304 (92.97%)	90 (95.74%)	214 (91.85%)	
Yes	23 (7.03%)	4 (4.26%)	19 (8.15%)	
Drinking, No (%)				0.3650
No	311 (95.11%)	91 (96.81%)	220 (94.42%)	
Yes	16 (4.89%)	3 (3.19%)	13 (5.58%)	
Insomnia, No (%)				0.1619
No	278 (85.02%)	84 (89.36%)	194 (83.26%)	
Yes	49 (14.98%)	10 (10.64%)	39 (16.74%)	
Hereditary, No (%)				0.8362
No	263 (80.43%)	78 (82.98%)	187 (80.26%)	
Yes	64 (19.57%)	18 (19.15%)	46 (19.74%)	

Statistics displayed as mean ± SD for continuous variables and N (%) for categorical variables. For continuous variables, comparisons were made using parametric paired *t*-tests or non-parametric Mann–Whitney tests, as appropriate. Statistical tests for categorical variables were performed using the Chi-square test. *p*-values < 0.05 were considered significant. VASI, vitiligo area score index; VIDA, vitiligo disease activity score.

## Data Availability

Data are available within the article and its Appendix A. The datasets were derived from sources in the public domain: Gene Expression Omnibus (GEO, https://www.ncbi.nlm.nih.gov/geo/ (accessed on 1 February 2022)).

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
