# Peer review of "1,25-Dihydroxyvitamin D3 Provides Benefits in Vitiligo Based on Modulation of CD8+ T Cell Glycolysis and Function"

_nutrients, 2023, doi:10.3390/nu15214697_

Round 1

Reviewer 1 Report

Comments and Suggestions for Authors

I've had the opportunity to review your manuscript detailing the effects of 1,25(OH)2D3 on vitiligo, specifically focusing on CD8+ T cell metabolism and function. Your work sheds light on an important aspect of vitiligo research, and the potential therapeutic implications of vitamin D are noteworthy. The experimental design, particularly the use of the MAV mouse model, is commendable.

However, I have provided several comments and suggestions aimed at enhancing the clarity, depth, and overall impact of the manuscript. It's essential to present your data in a clear and logical manner, ensuring that all findings are supported by appropriate controls and thorough statistical analyses. I believe that addressing these comments will further strengthen your manuscript and make it an even more valuable contribution to the field.

Detailed comments:

1. Title and Abstract:

  • Ensure the title accurately reflects the main findings and scope of the study.
  • The abstract should provide a clear, concise overview of the objectives, methodology, main findings, and significance of the study.

2. Introduction:

  • The authors should contextualize their work more clearly within the existing literature. They mention conflicting studies on vitamin D and vitiligo, but a more comprehensive review of previous findings might help set the stage for the present study.

3. Methodology:

  • The MAV mouse model: While the authors mentioned the MAV mouse model, they should provide more information about its origin, characteristics, and validation in the context of vitiligo research.
  • For the CCK-8 analysis, ensure that details such as incubation times, cell densities, and any relevant controls are mentioned. This will help in the reproducibility of the study.
  • More details on the statistical methods used, the software, and any relevant tests should be provided.

4. Results:

  • Presentation of Data: The results section should clearly and sequentially present the findings of the study. Ensure that the data is presented in a logical order that follows the methods section, allowing readers to easily follow the study's progression.
  • Figure Clarity: From the figures provided in our discussion, it's essential that each figure and its subparts have clear, readable labels, legends, and captions. This aids in ensuring that the reader can understand the content without frequently referring back to the text.
  • Statistical Significance: The annotations indicating statistical significance (e.g., p-values, asterisks) should be consistent across all figures. It would also be beneficial to include the exact p-values for all tests, rather than just significance markers. The nature of the statistical tests used should be mentioned in the figure captions or methods section.
  • Experimental Groups and Controls: Ensure that there is clear differentiation between experimental groups and controls in each presented dataset. The use of appropriate controls is essential to validate the findings, especially when assessing the effects of 1,25(OH)2D3.
  • Depth of Findings: The authors highlighted the effects of 1,25(OH)2D3 on CD8+ T cell metabolism and function, particularly its role in inhibiting glycolysis via the AMPK pathway. However, delving deeper into the broader implications of these findings is recommended. For instance, were there any observed effects on CD8+ T cell differentiation, other metabolic pathways, or other intracellular signaling pathways?
  • Inter-Experiment Consistency: For experiments that were replicated multiple times (e.g., cell culture assays), it would be valuable to know if there was consistency across replicates. Mention of any outlier data points or anomalous results would also provide a more comprehensive view of the data.
  • Comparative Analysis: When discussing the effects of varying doses of 1,25(OH)2D3, it would be beneficial to provide more detailed comparative analyses. For instance, the effects seen at the 5ug/kg dose compared to the 1ug/kg dose in the MAV mouse model.
  • Interpretation vs. Presentation: The results section should aim to present the data without extensive interpretation. Conclusions drawn from the data should be saved for the discussion section. Ensure that the results are not over-interpreted in this section but are presented in a straightforward manner.

5. Discussion:

  • The focus on the AMPK pathway and glycolysis in CD8+ T cells is interesting. However, it would be helpful to discuss the broader implications of these findings for other autoimmune or inflammatory conditions.
  • The potential off-target effects of 1,25(OH)2D3 should be discussed, especially since it is a bioactive molecule with roles in various physiological processes.
  • The authors mention that 1,25(OH)2D3 might be a "safe and economical treatment in vitiligo." This claim requires further discussion, especially regarding potential side effects, dosage considerations, and interactions with other treatments.

Reviewer 2 Report

Comments and Suggestions for Authors

I suggest making a longer introduction with more details from previous studies, even if they are conflicting.

Methods are well described but could help to understand with 2 figures, one for the in vitro model and the other for the animal model.

Any statistical calculation was made to ensure the number of samples was enough?

I suggest including a group without vitiligo and normal serum 1,25(OH)2D3 concentration.

It is important to know if the serum 1,25(OH)2D3 concentration was tested in the same laboratory and during the same period of the year for all subjects. Please include in the table the meaning of VASI and VIDA. Also, consider standardizing decimal numbers into 3 or 4 numbers for p value.

Please include legends of abreviations in all figures.

Reviewer 3 Report

Comments and Suggestions for Authors

I think that in the summary the context in which the study is developed should be clear.

I also believe that the number of subjects, the number of patients, should be reflected in the summary.

Perhaps the authors should think about whether the presentation of the material and method section group the sections into Subjects, procedure (and in this section include points 2.2 to 2.11), data sources and analysis, but it is only a suggestion

2. Materials and methods 74

2.1. Subjects

2.2. Cell culture and treatment

23. Cell viability

2.4. Glucose absorption assay

2.5. Enzymatic activity assays

2.6. Animal and drug treatment protocols

2.7. Enzyme-linked immunosorbent assay (ELISA

2.8. Flow cytometry analysis

2.9. Immunohistochemistry

2.10. Quantitative real-time polymerase chain reaction (qRT-PCR)

2.11. Western blot analysis

2.12. Data source and bioinformatic analysis.

2.13. Data presentation and statistical analysis.

The authors say

"Statistical analysis and graphical visualizations were performed in GraphPad Prism.

(v.8.1.0). Statistical analyzes were performed using appropriate paired or unpaired parameters.

metric and non-parametric tests, as necessary".

Presentation of results is good

Discussions are good

The conclusions are good

Reviewer 4 Report

Comments and Suggestions for Authors 1. I do not think that the title is accurate. 2. The sentences to 56 through 259 need to be reviewed. I do not think there are clear. The vitamin D concentration in line 258 is not apparent in table 1. 3. The vitamin D role in table 1 says 25 hydroxy vitamin D rather than 1–25 hydroxy vitamin D. 4. Most readers will not be as familiar with the vitiligo classifications used. These should be described briefly in the methods. Thanks

Comments on the Quality of English Language 1. I do not think that the title is accurate. 2. The sentences to 56 through 259 need to be reviewed. I do not think there are clear. The vitamin D concentration in line 258 is not apparent in table 1. 3. The vitamin D role in table 1 says 25 hydroxy vitamin D rather than 1–25 hydroxy vitamin D. 4. Most readers will not be as familiar with the vitiligo classifications used. These should be described briefly in the methods. Thanks
